# Electrochemical System for Field Control of Hg^2+^ Concentration in Wastewater Samples

**DOI:** 10.3390/s23031084

**Published:** 2023-01-17

**Authors:** Anda-Gabriela Tenea, Cristina Dinu, George-Octavian Buica, Gabriela-Geanina Vasile

**Affiliations:** 1National Research and Development Institute for Industrial Ecology ECOIND, 57-73 Drumul Podul Dambovitei Str., 060652 Bucharest, Romania; 2Chemistry Department, Science Faculty, University of Craiova, 107i Bucharest Street, 200478 Craiova, Romania; 3Faculty of Chemical Engineering and Biotechnologies, University Politehnica of Bucharest, 1-7 Polizu Str., 011061 Bucharest, Romania

**Keywords:** carbon screen-printed modified electrode, mercury, electrochemical detection, wastewater analysis

## Abstract

The paper presents the validation of an electrochemical procedure for on-site Hg^2+^ ions determination in wastewater samples using a modified carbon screen-printed electrode (SPE) with a complexing polymeric film based on poly(2,2′-(ethane-1,2-diylbis((2-(azulen-2-ylamino)-2-oxoethyl)azanediyl))diacetic acid) (poly**L**). Using metal ions accumulation in an open circuit followed by anodic stripping voltammetry, the SPE-poly**L** electrode presents a linear range in the range of 20 µg/L to 150 µg/L, with a limit of detection (LOD) = 6 µg/L, limit of quantification (LOQ) = 20 µg/L, and an average measurement uncertainty of 26% of mercury ions. The results obtained in situ and in the laboratory using the SPE-poly**L** modified electrode were compared with those obtained by the atomic absorption spectrometry coupled with the cold vapor generation standardized method, with the average values indicating excellent recovery yields.

## 1. Introduction

The toxicity of heavy metals imposes increasing concerns because of the effects they produce on human health and the ecological system. Mercury is one of the most toxic metals and at the same time one of the most widespread pollutants in nature, coming from natural and anthropogenic sources [1]. It is an element that does not biodegrade in the environment and is well known for its accumulation in the aquatic ecosystem. This element from the atmosphere contaminates waters (rivers, lakes, oceans) and soils. In the aquatic environment, inorganic mercury is transformed into organic mercury (methylmercury—the most toxic form). Populations that have a diet based especially on seafood or that consumes certain species of fish (king mackerel, shark, tuna, etc.) also have an increased risk of methylmercury poisoning.

Mercury (Hg^2+^) and its compounds are extremely toxic to the environment. Due to its accumulation in food, it is extremely harmful even at very low amounts. In addition, due to its strong absorption into biological tissues, it exhibits a very slow rate of elimination. Through this bioaccumulation process, mercury easily passes from the aquatic system into the food chain. Exposure and consumption of this element, even at very low concentrations, can cause neurological and movement disorders, kidney failure, cancer, liver dysfunction, and disorders of the endocrine systems [2]. According to the Environmental Protection Agency (EPA), the maximum allowed limit for Hg^2+^ is less than 10 nM [3]; therefore, monitoring environmental contamination with Hg^2+^ is of great interest.

Because of the threats related to mercury pollution, a strict control of this element in the environment is necessary, so it follows that the development of quick methods for detecting mercury in the field is also a strict necessity. Although there are several traditional analytical methods, which show great sensitivity and selectivity for its determination, most of them are laborious, voluminous, expensive, and time-consuming. For Hg^2+^ ions detection, methods such as colorimetry [4], fluorimetry [5], electrochemistry [6], and surface plasmon resonance [7] are used. The most used methods for mercury detection are atomic absorption spectrometry (AAS) with hydride generation, AAS with cold vapors, atomic fluorescence spectrometry, optical emission spectrometry with inductively coupled plasma, and mass spectrometry with inductively coupled plasma [8], but these methods also present some disadvantages, such as the need for a large sample volume, high costs, a long time required for sample pretreatment, and long test duration, which are labor-intensive and requires massive instruments and, therefore, makes their use in on-site and real-time mercury detection impossible. An alternative to traditional analytical methods for the detection of mercury ions is represented by electroanalytical methods based on chemically modified electrodes with suitable characteristics, such as low cost, simplicity of use, high selectivity, and sensitivity, were developed. Moreover, they can be included in portable devices, allowing in situ monitoring of analyzed samples [9]. An inconvenience of the electroanalytical technique is the use of unmodified electrochemical sensors because they have low selectivity, high overpotential, low sensitivity, and lack of reliability [10]. In order to improve the performance of electrochemical sensors for the detection of mercury at the maximum admitted level, certain materials are used such as graphene, nanoparticles, carbon nanotubes, or organo-metallic compounds [11,12,13,14].

For the determination of toxic metal ions in environmental samples, their on-site quantification is an essential and mandatory process in the industry and water treatment processes. Sophisticated instruments or devices cannot be used for in situ detection of metal ions, which is why their analysis using disposable sensors (SPE) is the most widely used in the field of electroanalysis [15,16]. In order to increase the sensitivity and selectivity towards certain analytes, various carbon-based electrode materials were used by modifying their surfaces with substances that have a selective affinity towards the metal ions of interest [17,18]. Several types of carbon materials, including glassy carbon, graphene, carbon nanotubes, carbon paste, carbon fibers, and SPEs, have been reported as substrate materials for the detection of metal ions [19,20,21]. In recent years, screen-printing has been the most used in electroanalysis for the manufacture of sensors and biosensors [22,23,24]. SPEs usually include three electrodes (the working electrode, the counter electrode, and the reference electrode) and have attractive features such as easy to manufacture, low cost, and small sample volume (µL) [25].

Higher selectivity and sensitivity compared to other methods [26,27], low cost, and simplicity of use are a few characteristics of electrochemical sensors based on chemically modified electrodes. These sensors can be included in portable devices, allowing in situ monitoring of the analyzed samples [28]. Recently, a group of researchers from China developed an electrochemical method for the rapid detection of methylmercury in water samples: a method that requires a short sample preparation time and low cost. The developed electrochemical sensor used gold nanoparticles and a glassy carbon electrode modified with zeolitic imidazolate-67, obtaining through this technique a detection limit of 0.05 µg/L [29]. Chemically modified electrodes based on polymeric films were also tested for heavy metals monitoring in water samples [30,31]. Particularly, azulene-based sensors were tested for some heavy metals detection, obtaining adequate results [28]. Therefore, their employment for mercury monitoring may be promoted. Moreover, for the in situ detection of mercury ions, a series of nanomaterials with intrinsic catalytic activity, called nanozymes, were used. These nanozymes show a behavior similar to that of native enzymes, which is why they were applied in the detection of mercury based on color variation [32].

Recently, nanoparticles have been used to remediate soils and waters contaminated with metals. These nanoparticles are used due to their large surface area, high adsorption capacity, fast diffusion speed, and significant chemical reactivity [33,34,35]. In order to prevent environmental pollution with mercury, strategies based on nanomaterials have been developed using graphene oxide, carbon nanotubes, or carboxymethyl cellulose or chitosan as adsorbent material to remove mercury from polluted water [36].

Electroanalytical methods are an alternative solution to the traditional methods of mercury detection in water samples. Despite the fact that numerous studies have been published based on the use of disposable SPEs for monitoring metal contaminants in the environment [37,38], there are only a few studies that employed SPEs to detect mercury levels in the environment [39]. For instance, Joe Wang was the first to employ a SPE gold-based electrode for potentiometric stripping measurements of trace mercury levels, achieving a detection limit of 0.5 µg/L [40]. In order to detect mercury, Bernalte et al. in 2011 employed a commercially available carbon-based SPE electrode enhanced with gold nanoparticles, obtaining a LOD of 1.1 µg/L for testing on samples of water and dust using square-wave anodic stripping voltammetry [41,42,43]. In addition, electrodeposited gold on a SPE was reported by Mandil et al. for mercury detection, achieving a LOD of 1.5 µg/L [44].

This study aimed to validate the electrochemical method for the determination of mercury from wastewater samples using a modified SPE with complexing polymeric film based on poly**L** (the structure of the monomer is presented in Figure 1.). In addition, the present work intends to make a step further into the development of an electrochemical portable device, with low running costs and easy-to-operate system based on a modified SPE for mercury detection. 

## 2. Materials and Methods

### 2.1. Reagents

The monomer 2,2′-(ethane-1,2-diylbis((2-(azulen-2-ylamino)-2-oxoethylazanediyl)diacetic acid (**L**) (Figure 1) was used to modify the carbon SPE through oxidative polymerization under imposed potential as previously shown [45]. 

Acetonitrile (Sigma Aldrich, Saint Louis, MO, USA, purity 99.999%), tetrabutyl ammonium perchlorate—TBAP (Sigma Aldrich, Saint Louis, MO, USA, for electrochemical analyses, ≥99.0%), anhydrous glacial acetic acid (Supelco, Saint Louis, MO, USA, 100%), anhydrous sodium acetate (Merck, Darmstadt, Germany, 99.99%), and Hg^2+^ standard solution, mercury acetate ((CH_3_CO_2_)_2_Hg) (Fischer Chemical, Waltham, MA, USA, quality pa) were used as received. 

The reagents used for alternative method were certified reference material, 1001 mg/L Hg in 10% HNO_3_ (CPAChem, Stara Zagora, Bulgaria), tin (II) chloride dihydrate for analysis (Supelco, Saint Louis, MO, USA), potassium bromide puriss (Sigma Aldrich, Saint Louis, MO, USA), potassium bromate puriss (Sigma Aldrich, Saint Louis, MO, USA), hydrochloric acid 34–37% for trace metal analysis (VWR Chemicals, Radnor, PA, USA), nitric acid suprapur 69% (Supelco, Saint Louis, MO, USA). All solutions were prepared using ultrapure water 18.2 MΩ·cm.

### 2.2. Equipment

The SPEs were purchased from BVT Technologies (Strazek, Czech Republic). The characteristics of these electrodes are: carbon working electrode (2 mm diameter); auxiliary carbon electrode; Ag/AgCl reference electrode. The dimensions of these electrodes were mass: 0.5 g; length: 25.40 mm; width: 7.26 mm; thickness: 0.63 mm.

PalmSens4 portable potentiostat connected to a laptop and equipped with PSTrace software, capable of processing all the information provided by the potentiostat from the electrochemical cell (EC) and rendering it in the form of graphics on a computer screen. The connection between the EC and the potentiostat was made with the help of a connector from Metrohm DropSens (reference CAC4MMH).

A Thermo M6 spectrometer equipped with cold vapors and a Hg cavity cathode lamp was used to perform the atomic absorption spectrometry with cold vapor generation (AAS-CV) standardized.

A Millipore Direct-Q water purification system from Merck was used a source of purified water.

### 2.3. Procedures

#### 2.3.1. Electrochemical Procedures for Hg^2+^ Analysis

The SPE modified electrode was prepared in supporting electrolyte (0.1 M) obtained from acetonitrile and TBAP containing 1.5 mM **L** using the previously described procedure [45]. An example of successive cyclic voltammograms of **L** on SPE is presented in Appendix A, which emphasize an accumulation of polymeric material on the electrode surface during electropolymerization. After its transfer into free monomer electrolyte (Appendix A), the SPE-poly**L** modified electrode maintains its electroactivity which confirms the electrodeposition of the complexing polymeric film on these electrodes. The aqueous acetic acid buffer (0.2 M) was obtained from anhydrous glacial acetic acid and anhydrous sodium acetate. For mercury ions detection, an electrochemical measuring cell was used (with acetate buffer concentration 0.1 M) and an ion complexation cell of Hg^2+^ (Figure 2) at pH = 3.

Mercury standards solutions were obtained from mercury acetate in acetate buffer (pH = 3.0). The calibration curve was in the concentration range 20–150 µg/L (20 µg/L; 40 µg/L; 50 µg/L; 60 µg/L; 80 µg/L; 100 µg/L; 150 µg/L). The analytical measurements were carried out at a pH = 3 of the acetate buffer solution both in the electrochemical measurement cell and in the Hg^2+^ ions complexation cell. The Hg^2+^ ions detection experiments were carried out in acetate buffer (0.1 M or 0.05 M) at pH = 3. The detection procedure consists of the following successive stages:SPE electrodes modified with poly**L** films (SPE-poly**L**) are inserted into acetate buffer (0.1 M, pH = 3) and a cyclic voltammetry (CV) between −0.5 V and +1.2 V is performed. In this way, the electroactivity of the polymer film is destroyed, obtaining a lower background current. Before analysis, a differential pulse voltammogram in free analyte solution was recorded to observe the background current. Conditions: DPV: equilibration time 5 s; starting potential 0.5 V; potential vertex 1 −1.3 V; potential vertex 1 0.5 V; potential step 0.01 V; scanning speed 0.024 V/s, number of cycles 1.After stage I, the SPE-poly**L** electrodes are immersed in 20 mL complexation solution (Hg^2+^ ions in acetate buffer (0.05 M, pH = 3), kept for 20 min under stirring (open circuit complexation).After stage II, the SPE-poly**L** electrodes are removed from the complexation solution and rinsed with ultrapure water to remove traces of uncomplexed Hg^2+^ ions.The electrodes are inserted into the acetate buffer (0.1 M, pH = 3) where the Hg^2+^ ions retained on the surface of the SPE-poly**L** electrodes are reduced to zero valence by polarizing the electrodes at −1.3 V for 15 s followed by their reoxidation using DPV under the following conditions: conditioning potential 0 V; conditioning time 0 s; storage time 15 s; storage potential −1.3 V; equilibration time 5 s; start potential −1.3 V; closing potential 1 V; potential step −0.005 V, pulse time 0.05 V; scan speed 0.01 V/s. 

This four-stage procedure was applied for the validation of the proposed method for Hg^2+^ detection (Appendix A), using at this level only standard solutions from mercuric acetate.

To validate the electrochemical method, the following performance parameters of the method were verified: linearity and working range, LOD and LOQ, accuracy, repeatability, intermediate precision, recovery, robustness, and measurement uncertainty. 

#### 2.3.2. Alternative Method for Control of Hg^2+^ in Wastewater Samples

An alternative method used for quality control of the Hg^2+^ ions in wastewater samples, in order to validate the electrochemical procedures applied in the study, was the EN ISO 12846:2012 standardized method [46]. The technique of this standardized method is AAS-CV. 

#### 2.3.3. Wastewater Sampling 

Seventeen wastewater samples were collected from an industrial site in order to determine mercury. Duplicate samples were preserved with KBrO_3_-KBr in hydrochloric acid and transported to the analytical laboratory for mercury determination with the AAS-CV technique. In addition, electrochemical procedures were applied in the laboratory using PalmSens4 portable potentiostat equipment. On-site, the electrochemical procedures were performed and the concentration of Hg^2+^ ions was determined using the same device. 

## 3. Results

### 3.1. Electrochemical Performance of SPE-poliL for Hg^2+^ Analysis

#### 3.1.1. Linearity and Working Range

To establish the linearity of the method, the calibration curve was performed on six points in the concentration range from 20 to 150 µg/L Hg^2+^ (20 µg/L; 50 µg/L; 60 µg/L; 80 µg/L; 100 µg/L; 150 µg/L). For each concentration level, three replicates were analyzed; statistical parameters were calculated using the average of these three replicates. Good correlation coefficient (R = 0.9996) and determination coefficient (R^2^ = 0.9993) values were obtained, with the values being close to 1.

The linearity of the regression curve calculated with Equation (1) indicated a value of 99.07%, which falls within the range of ±1% [47]. The value of the method coefficient of variation (1.23%) was less than 2% according to Horwitz function, situating this approach as a good electrochemical method [48].
Linearity = (1 − s_b_/b) × 100(1)

Figure 3 presents the calibration curve that was used to establish the performance parameters of the method.

#### 3.1.2. LOD, LOQ

For LOD and LOQ, six replicates enriched with 20 µg/L Hg^2+^ solution were prepared from mercury acetate. The average value of the replicates was X average = 21.9 µg/L. For estimation of LOD and LOQ, Equations (2) and (3) were used.
LOD = 3 × s(2)
LOQ = 10 × s(3)

The standard deviation of the replicates (s = 1.9747 µg/L) was obtained and, according to Equations (2) and (3), 6.0 µg/L (LOD) and 20.0 µg/L (LOQ) were reported.

#### 3.1.3. Accuracy

To determine the accuracy of the method, ten individual replicates were performed at five concentration levels (20 µg/L Hg^2+^, 50 µg/L Hg^2+^, 80 µg/L Hg^2+^, 100 µg/L Hg^2+^, 150 µg/L Hg^2+^). The replicates were carried out by a single analyst, in the same day using the same equipment and the same working conditions and method. The obtained results are presented in Table 1.

#### 3.1.4. Repeatability

The repeatability of the method was carried out on ten individual replicates having the concentrations 20 µg/L Hg^2+^, 50 µg/L Hg^2+^, 80 µg/L Hg^2+^, 100 µg/L Hg^2+^, and 150 µg/L Hg^2+^. The replicates were realized by a single analyst on the same day using the same equipment and the same work method. The obtained results (standard deviation of the repeatability, repeatability, and the repeatability variation coefficient) are presented in Table 2.

The repeatability and repeatability variation coefficient are presented in the following equations:*Repeatability (r) = 2.8 × S_r,_(4)
**RSDr = (S_r_ × 100)/X_average_(5)

#### 3.1.5. Intermediate Precision

To determine the intermediate precision (R), twelve replicates were analyzed at three concentration levels (20 µg/L, 50 µg/L, and 100 µg/L Hg^2+^). The tests were carried out by two analysts in three days using the same equipment and the same working condition and method. The obtained results (average concentration, standard deviation of the intermediate precision (S_R_), intermediate precision, and standard deviation of the intermediate precision RSD_R_) are presented in Table 3.

The intermediate precision and intermediate precision variation coefficient are presented in the following equations:*Intermediate precision (R) = 2.8 × S_R,_(6)
**RSDR = (S_R_ × 100)/X_average_(7)

#### 3.1.6. Interference

The selectivity of the poly**L** material was checked and discussed in our previous paper [45]. In addition, we can say that, as can be seen from Appendix A, the mercury ions stripping peak current remained at 90% from the initial height even when five mass equivalents of interference ions I (I: (Zn(II), Cd(II), Pb(II), Ni(II), Co(III), and Cu(II)) were added.

#### 3.1.7. Recovery

The recovery test indicates, according to the literature, “the fraction of the analyte added to the test sample (fortified or spiked) prior to analysis which is measured by the method” [48].

Recovery percentages were calculated after analyzing six replicates of standard solutions of 50 µg/L and 100 µg/L. Same standard solutions was added to real samples without Hg^2+^ (spike matrix sample) and six replicates with same standard solutions added to the blanks (spike blank) with the following equation:Recovery = (X_spike matrix sample_ × 100)/X_spike blank_,(8)

Recovery test data indicated recovery percentage situated in the range of 98.2% ± 5% (100 µg/L) to 110% ± 7% (50 µg/L). 

#### 3.1.8. Robustness

The robustness tests were performed using the Youden and Steiner partial factorial model [49,50]; a model in which three factors can be varied and analyzed with only four experiments. This efficient method can be used for “in-house” validation of the methods. Depending on the procedure used, a high value (+) and a low value (−) are established for each factor, varying by a maximum of 10% the normal value of the factor. The partial factorial model for three factors is presented in Table 4.

The absolute effect (bias) of each factor from A to C can be evaluated with the following formula:Effect _A_ (absolute value) = │(ΣY_A+_ − ΣY_A−_)/2│(9)
where: $\sum Y_{A+}$ = sum of Y results, where factor A has the high value (+), Y_1_ + Y_3_, n = 2; $\sum Y_{A-}$ = sum of Y results, where the factor A has the low value (−), Y_2_ + Y_4_, n = 2.

The effect of a factor can be considered significant if it exceeds the value of 1.4 multiplied by S_R_, where S_R_ is the standard deviation of the intermediate precision value, according to Equation (10): Effect _A÷C_ > 1.4 × S_r_(10)

This formula could be used in the case of the new method’s “in-house” validation. S_R_ could be replaced with other value, such as standard deviation of the original procedure taken from the last control test.

The relative effects of each factor can be estimated using the following formula:Effect _A_(%) = [(ΣY_A+_ − ΣY_A−_) × 100]/ΣY_A+_
(11)

The factors selected for testing the robustness of the electrochemical method for the determination of Hg^2+^ using DPV technique with the SPE-poly**L** modified electrodes were the parameters that influence the retention of Hg^2+^ ions on the surface of SPE-poly**L** electrodes during DPV procedures: pH (pH = 3), reaction time (20 min), acetate buffer concentration (0.05 M).

Each experiment was performed in six replicates so that the Y_i_ value of each experiment represented their average value. The tested concentration was 20 µg/L. 

Two robustness tests were performed: one in which all three parameters were modified by 10% compared to the normal value, and the other in which two parameters were modified by 10% (reaction time and acetate buffer concentration), and one was modified by 3.5% (pH value). Table 5 presents the values of the factors for each experiment.

Table 6 presents the values of Hg^2+^ concentration (µg/L) for both tests; values obtained when the pH, reaction time, and acetate buffer concentration were modified according to the data for A, B, and C factors (Table 5). 

Table 7 presents the data obtained in test 1 to verify the robustness of the electrochemical method. The ΣY_A+_ represents the sum of average values from T1-Y_1_ and T1-Y_3_: the experiments where Factor A had the highest value (pH = 3.3). The same ΣY_A−_ represents the sum of average values from T1-Y_2_ and T1-Y_4_: the experiments where Factor A had the lowest value (pH = 2.7). The ΣY_B+_ represent the sum of average values from T1-Y_1_ and T1-Y_2_: the experiments where Factor B had the highest value (reaction time = 22 min), while the ΣY_B−_ represents the sum of average values from T1-Y_3_ and T1-Y_4_: the experiments where Factor B had the lowest value (reaction time = 18 min). Regarding factor C, the highest value was 0.055 M acetate buffer concentration (the ΣY_C+_ was obtained from the sum of average values from T1-Y_1_ and T1-Y_3_) and the lowest value was 0.045 acetate buffer concentration (the ΣY_C−_ was obtained from the sum of average values from T1-Y_2_ and T1-Y_4_). Values for absolute effect and relative effect for each factor were calculated using Equations (9) and (11). 

The comparison value used, according to Equation (11), is 10.5 µg/L, with S_R_ for 20 µg/L being 7.52 µg/L (Table 3). 

Table 8 presents the data obtained in test 2 to verify the robustness of the method. The values of the factors and their influence on the method were calculated in the same way as in test 1.

#### 3.1.9. Uncertainty Budget

In the estimation of the measurement uncertainty, data from the quality certificates of the reagents, the analytical balance, graduated flasks, and pipettes used in the entire procedures, as well as data from the precision and recovery tests, were used. Table 9 presents the values of uncertainty at three different concentrations. 

### 3.2. Electrochemical Analysis of Hg^2+^ in Wastewater Samples

To verify the functionality of the electrochemical procedure with SPE modified with poly**L** films, mercury concentration from wastewater samples was determined in the laboratory and on-site using the portable potentiostat PalmSens4 connected to a laptop and equipped with PSTrace software. The obtained results were compared with those from the AAS-CV standardized method. Table 10 presents the values of Hg^2+^ concentrations in seventeen wastewater samples.

## 4. Discussion

Usually, electrochemical studies report only few descriptive statistics and metrics, such as the standard deviation of the results, and thus complex validations of the proposed methods are not carried out, both in the case of electrochemical methods used to determine metals and pharmaceutical compounds.

In our previous studies, we introduced statistics and metrics other than standard deviation in the experimental tests, such as repeatability, intermediate precision, and measurement uncertainty [51,52].

In the present study, full validation test for a new method was applied in order to offer a solution for Hg^2+^ ion monitoring in wastewater to the analytical laboratories that do not have expensive equipment and specialized staff for quality control of water.

Thus, from the linearity test it was observed that the variation coefficient of the method is 1.23%, a very good value according to the literature [47]. Regarding repeatability (r) and intermediate precision (R) of the analytical methods, the literature data indicate that at the level from LOQ to 2LOQ, repeatability must be less than 25% and intermediate precision around 30% [47]. Obtained values reported in Table 2 and Table 3 indicated that the r and R values fell within the accepted range. For concentration from 2LOQ to 10 LOQ, r must be lower than 15% and R must be lower than 20% [47]. Repeatability and intermediate precision expressed in terms of percentage for the concentrations between 50 µg/L and 150 µg/L (Table 2 and Table 3) were situated below the admissible values. 

Regarding recovery percentage, the literature indicates a recovery range between 80% and 110% for the concentration of 100 µg/L [47]: values in which the recovery tests fall.

From the data obtained in the robustness tests, it can be seen that the pH and the acetate buffer concentration do not significantly influence the electrochemical determination of Hg^2+^ using the DPV technique with the SPE-poly**L** modified electrodes. The absolute effects of factors A (pH) and C (acetate buffer concentration) in both tests had lower values than the comparison value of 10.5 µg/L (2.85 µg/L in test 1, 4.84 µg/L in test 2). In terms of relative effect, variation with ± 10% of factors A and C values indicates an influence from 15.4% (test 1) to 31.2% (test 2). The pH value situated in the range 2.7 to 3.3 pH units and acetate buffer concentration from 0.045 M to 0.05 M does not influence the electrochemical procedure.

Instead, in both tests, the absolute effect of factor B (reaction time) had a higher value than the comparison one (14.0 µg/L in test 1, 13.5 µg/L in test 2). In addition, a higher value of relative effect was obtained: 107% (test 1) to 210% (test 2).

The reaction time between the SPE-poly**L** modified electrode and the solution with Hg^2+^ ions is crucial for the method. The 10% variation of the reaction time indicates a major change; the values obtained in both tests (Table 7 and Table 8) show that the change of this factor determines a lack of the method robustness. 

The “in-house” validated method for the determination of Hg using modified SPE with complexing polymeric film based on poly**L** leads to precise and accurate results, with a measurement uncertainty of 26% at 50 µg/L Hg^2+^, which represents the maximum allowed limit for wastewater originating from treatment plants and discharged into natural receptors. For these reasons, the method is suitable for controlling and monitoring on-site of Hg^2+^ ions in wastewater samples. 

The data presented in Table 10 regarding Hg^2+^ concentration in wastewater samples show comparable results both in the field and in the laboratory using the electrochemical procedure with SPE-poly**L** modified electrodes. In addition, the electrochemical results (EC lab, EC on-site) were in the same range with those obtained from the AAS-CV technique. 

## 5. Conclusions

The validated method allows the determination of the Hg^2+^ concentration directly at the pollution source using a method with low complexity and affordable equipment.

The proposed procedure for the determination of Hg^2+^ using the SPE-poly**L** modified electrode has a wide linear range, low LOD and LOQ values, suitable repeatability, intermediate precision, and uncertainty values. According to the robustness tests, pH and acetate buffer concentration does not influence the method in the range studied (+10% variation), while reaction time has a strong effect in the procedure. Hence, the last parameter should be carefully considered to obtain adequate results. The results obtained in situ and in the laboratory were compared with those obtained using the standardized method, with the average values indicating very good recovery yields and the Hg^2+^ concentrations being both in the linear range of the calibration curve and below the quantification limit. The electrochemical procedure for the determination of Hg^2+^ using the SPE-poly**L** electrode is suitable for field monitoring of wastewater. 

## 6. Patents

Invention patent application OSIM (Romanian State Office for Patents and Brands) no. A00464/29.07.2022. A method for obtaining a screen-printed electrode modified with polymeric films and an electrochemical procedure for field determination of mercury concentration in wastewater.

## Figures and Tables

**Figure 1 sensors-23-01084-f001:**
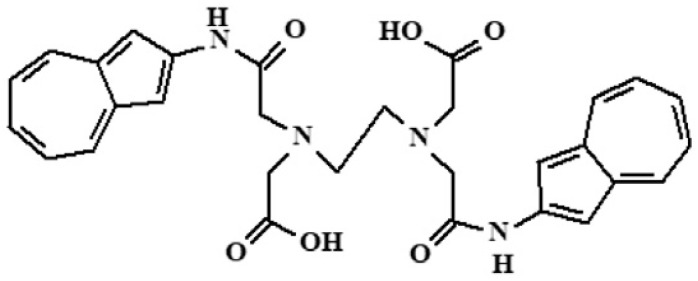
The structure of (2,2′-(ethane-1,2-diylbis((2-(azulen-2-ylamino)-2-oxoethyl)azanediyl))diacetic acid, (**L**).

**Figure 2 sensors-23-01084-f002:**
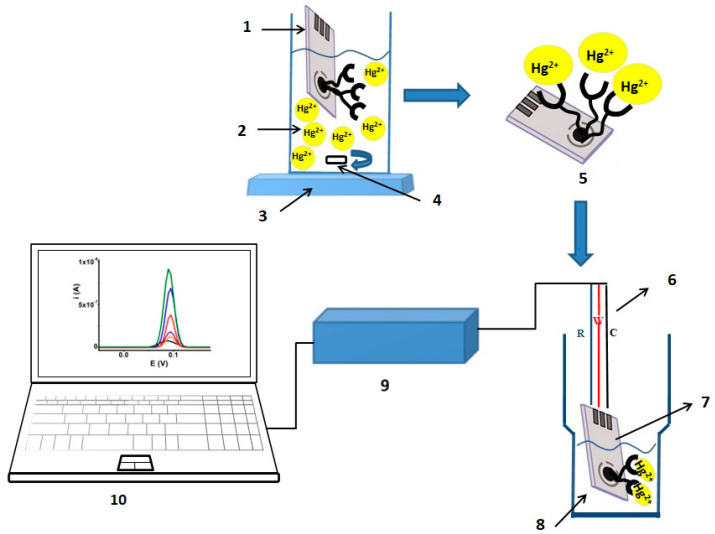
Principle scheme for the realization of the electrochemical process for the determination of Hg^2+^ ions in wastewater: 1—electrode with deposited polymer; 2—electrochemical cell with 20 mL 0.05 M acetate buffer solution at pH = 3; 3—magnetic stirrer; 4—Teflon magnet; 5—electrode with complexed Hg^2+^ ions and uncomplexed Hg^2+^ ions; 6—three electrical contacts; 7—electrode with complexed Hg^2+^ ions; 8—electrochemical cell; 9—portable potentiostat; 10—laptop with installed software.

**Figure 3 sensors-23-01084-f003:**
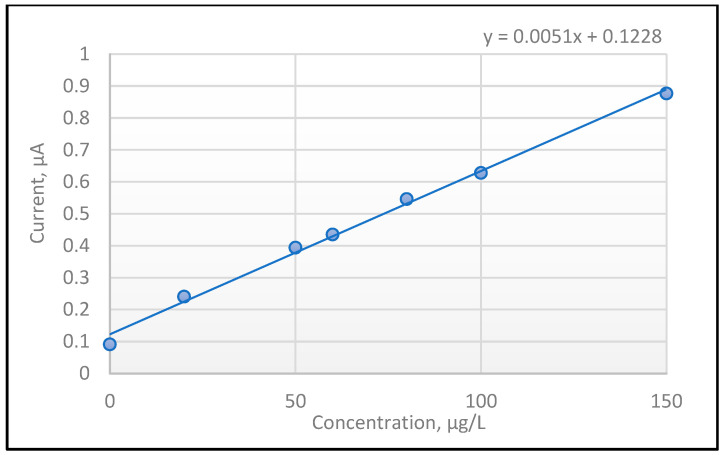
Linear regression curve for Hg(II) through the DPV technique with SPE-poly**L**.

**Table 1 sensors-23-01084-t001:** Accuracy test coefficients.

Added Concentration (µg/L)	X_average_ (µg/L) ± s (Standard Deviation)
20	22.9 ± 3.44
50	53.2 ± 7.52
80	79.7 ± 10.6
100	100 ± 12.6
150	147 ± 13.0

**Table 2 sensors-23-01084-t002:** Repeatability test coefficients.

Added Concentration (µg/L)	X_average_ (µg/L)	S_r_(µg/L)	r* (µg/L)	RSDr** (%)
20	22.9	3.44	9.64	15.0
50	53.2	7.52	21.1	14.1
80	79.7	10.6	29.7	13.3
100	100	12.6	35.2	12.6
150	147	13.0	36.3	8.80

**Table 3 sensors-23-01084-t003:** Intermediate precision test coefficients.

Concentration/Parameter Determined	20 µg/L	50 µg/L	100 µg/L
X_average_, µg/L	24.2	55.1	98.2
S_R_, µg/L	7.52	10.0	12.3
R*, µg/L	21.1	20.9	34.6
RSD_R_**, %	28.1	18.2	12.6

**Table 4 sensors-23-01084-t004:** The partial factorial model for testing the robustness of the analytical method.

Experiment	Factor			Result
	A	B	C	
1	+	+	+	Y_1_
2	−	+	−	Y_2_
3	+	−	+	Y_3_
4	−	−	−	Y_4_

**Table 5 sensors-23-01084-t005:** The values of the factors for robustness tests.

**TEST 1**	**Factor**			**Result**
	**pH (A)**	**Reaction Time, Minutes (B)**	**Acetate Buffer Concentration (C)**	
Experiment 1	3.3	22	0.055	T1-Y_1_
Experiment 2	2.7	22	0.045	T1-Y_2_
Experiment 3	3.3	18	0.055	T1-Y_3_
Experiment 4	2.7	18	0.045	T1-Y_4_
**TEST 2**	**Factor**			**Result**
	**pH (A)**	**Reaction Time, Minutes (B)**	**Acetate Buffer Concentration (C)**	
Experiment 1	3.1	22	0.055	T2-Y_1_
Experiment 2	2.9	22	0.045	T2-Y_2_
Experiment 3	3.1	18	0.055	T2-Y_3_
Experiment 4	2.9	18	0.045	T2-Y_4_

**Table 6 sensors-23-01084-t006:** Hg^2+^ concentrations (µg/L) in test 1 and test 2 for all experiments, six replicates for each experiment.

Experiment	T1-Y_1_	T1-Y_2_	T1-Y_3_	T1-Y_4_	T2-Y_1_	T2-Y_2_	T2-Y_3_	T2-Y_4_
Replicate 1	7.75	18.1	23.7	24.4	5.04	4.82	20.0	12.5
Replicate 2	10.1	12.1	21.3	27.3	6.37	7.56	29.2	16.9
Replicate 3	8.35	21.4	26.8	29.6	8.47	8.27	26.3	14.3
Replicate 4	9.8	15.5	29.6	23.0	5.51	5.47	19.5	19.5
Replicate 5	11.7	15.0	24.6	30.1	5.28	4.69	26.9	11.8
Replicate 6	9.78	16.0	39.3	24.2	9.29	6.04	24.8	16.8
Average	9.60	16.4	27.5	26.4	6.66	6.14	24.4	15.3

**Table 7 sensors-23-01084-t007:** Quantification of the variability of the operating parameters of the Hg^2+^ determination procedure using DPV technique with SPE-poly**L** modified electrodes, Test 1.

Estimated Parameter	UM	Comparison Value	Quantifying the Influence of Factors					
			Factor A		Factor B		Factor C	
			ΣY_A+_	ΣY_A−_	ΣY_B+_	ΣY_B−_	ΣY_C+_	ΣY_C−_
ΣY_F+_/ΣY_F−_	µg/L	-	37.1	42.8	26.0	53.9	37.1	42.8
Absolute effect	µg/L	10.5	2.85		14.0		2.85	
Relative effect	%	-	15.2		108		15.2	

**Table 8 sensors-23-01084-t008:** Quantification of the variability of the operating parameters of the Hg^2+^ determination procedure using DPV technique with SPE-poly**L** modified electrodes, Test 2.

Estimated Parameter	UM	Comparison Value	Quantifying the Influence of Factors					
			Factor A		Factor B		Factor C	
			ΣY_A+_	ΣY_A−_	ΣY_B+_	ΣY_B−_	ΣY_C+_	ΣY_C−_
ΣY_F+_/ΣY_F−_	µg/L	-	31.1	21.4	12.8	39.7	31.1	21.4
Absolute effect	µg/L	10.5	4.84		13.5		4.84	
Relative effect	%	-	31.1		210		31.1	

**Table 9 sensors-23-01084-t009:** Uncertainty values for electrochemical method.

Concentration, µg/L	Value ± Uncertainty, µg/L	Uncertainty, %
25	25.1 ± 7.80	31.1
50	56.3 ± 14.7	26.0
90	86.3 ± 17.6	20.4

**Table 10 sensors-23-01084-t010:** Hg^2+^ concentration in wastewater samples using the electrochemical procedure with SPE-poly**L** modified electrodes and AAS-CV.

**Method**	**P1**	**P2**	**P3**	**P4**	**P5**	**P6**	**P7**	**P8**	**P9**
AAS-CV	45.3 ± 4.08	20.2 ± 1.82	35.3 ± 2.19	52.1 ± 4.69	68.6 ± 6.17	78.3 ± 7.05	23.2 ± 2.09	74.9 ± 6.74	30.0 ± 2.70
EC lab	44.1 ± 8.82	18.4 ± 3.69 *	35.8 ± 7.16	58.7 ± 11.8	66.0 ± 13.2	75.9 ± 15.2	23.8 ± 4.76	77.4 ± 15.5	23.9 ± 4.77
EC on-site	44.4 ± 8.89	20.2 ± 4.04	34.4 ± 6.89	58.2 ± 13.5	67.6 ± 6.17	75.9 ± 15.2	22.4 ± 4.48	78.4 ± 15.2	31.7 ± 6.35
**Sample**	**P10**	**P11**	**P12**	**P13**	**P14**	**P15**	**P16**	**P17**	
AAS-CV	21.8 ± 1.96	17.0 ± 1.53	36.1 ± 3.25	66.6 ± 5.99	21.9 ± 1.97	34.1 ± 3.07	50.8 ± 4.57	56.3 ± 5.07	
EC lab	18.9 ± 3.78 *	17.3 ± 3.45 *	34.4 ± 6.88	63.2 ± 12.7	20.0 ± 4.01	32.5 ± 6.50	48.4 ± 9.68	51.8 ± 10.4	
EC on-site	20.2 ± 4.04	18.2 ± 3.64 *	31.9 ± 6.39	61.8 ± 12.4	23.0 ± 4.61	31.4 ± 6.29	48.0 ± 9.61	52.3 ± 10.5	

* Informative values, below the LOQ (20 µg/L).

## Data Availability

RO-BOPI 11/29.11.2022, pp. 44, https://www.osim.ro/images/Publicatii/Inventii/2022/bopi_11_2022.pdf. (accessed on 21 December 2022).

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
