# Peer review of "Electrochemical System for Field Control of Hg2+ Concentration in Wastewater Samples"

_sensors, 2023, doi:10.3390/s23031084_

Round 1

Reviewer 1 Report

The paper reports the validation of an electrochemical procedure in wastewater samples by means of anodic stripping differential pulse voltammetry using a screen printed electrode modified with an azulene-based polymer. The electrochemical performance of the device for mecury detection was also studied through several electrochemical tests in buffer solution.

The research article is really interesting and provides suitable results with appropiate conclusions. Moreover, it proposes an alternative for mercury monitoring in wastewater from an electrochemical point of view, which it lies into the scope of the journal. For these reasons, I recommend its publication in this journal after considering some modifications in terms of the exposition of the results and some experiments which could be carried out.

Introduction

-At the beginning, ''mercury'' word was frequently used. To avoid repetitions, I think that other words, such as '' this element'', ''this metal'' or other ones can be employed.

-I understand that the use of SPE-modified materials for the detection of mercury were reported in lines 86-89, but the meaning is unclear. I believe that this part can be reorganized as follows: ''Several chemically modified electrodes with suitable characteristics, such as low-cost, simplicity of use, high selectivity and sensitivity were developed. Moreover, they can be included in portable devices, allowing in-situ monitoring of analyzed samples''.

-Some aspects concerning the use of polymers deposited on SPE for the monitorization of heavy metals could be mentioned after the exposition of screen printed sensors for mercury detection (lines 109-117) and before the purpose of the paper (lines 118-121). Afterwards, the employment of azulene-based sensors could be mentioned as a suitable option to monitorize mercury, based on some reports cited below in the paper. This is an example of what I mean: ‘’Chemically modified electrodes based on polymeric films were also tested for heavy metals monitoring in water samples. Particularly, azulene-based sensors were tested for some heavy metals detection, obtaining adequate results. Therefore, their employment for mercury monitoring may be promoted’’. This part of the paragraph can be linked with the one included in lines 118-121. Some research works should be cited in this part. Here I am including some examples after performing a brief literature search:

 *A high-performance electrochemical sensor for the determination of Pb(II) based on conductive dopamine polymer doped polypyrrole hydrogel, https://doi.org/10.1016/j.jelechem.2021.115815.

*Novel nano-engineered environmental sensor based on polymelamine/graphitic-carbon nitride nanohybrid material for sensitive and simultaneous monitoring of toxic heavy metals, https://doi.org/10.1016/j.jhazmat.2021.126267.

*Azulene-ethylenediaminetetraacetic acid: A versatile molecule for colorimetric and electrochemical sensors for metal ions, https://doi.org/10.1016/j.electacta.2018.01.059.

Experimental

-From my point of view, the experimental part is not well-organized. I think that it would be better to include all reagents within one subsection entitled ‘’Reagents’’, the instrumentation within another one entitled ‘’Instrumentation’’ and the procedures in different subsections (electrodeposition of poly-L on SPE, electrochemical analysis of Hg2+ in buffer and wastewater samples, sampling of wastewater, among others).

-The caption of the figure 1 refers the poly-L structure, but the figure includes the monomer (L). Moreover, I think that this figure can be moved to the end of the introduction section.

-I understand that DPV was employed to check if the electrochemical pretreatment was done successfully (lines 168-169). If I am right, it would be better to modify this part as follows: ''before analysis, a differential pulse voltammogram in free analyte solution was recorded''.

-The instrumental cyclic voltammetry conditions exposed in lines 170-172 are not clear. It would be more adequate to rewrite this sentence as follows: ''a cyclic voltammogram from -0.5 V to 1.2 V was recorded at 24 mV/s''. Hence, the last part of this paragraph (lines 170-172) could be removed.

-Consider to modify this expression ''using voltammetry in differential pulse '' (line 182) for this one: ''using differential pulse voltammetry (DPV) under the following conditions''. I also recommend to remove the expression ''Differential Pulse Voltammetry (DPV) conditions'' (lines 182-183).

Results

-I think that the results section is not well-presented. In my opinion, it could be divided into two major subsections: one of them focused on the analytical performance of the electrochemical device for Hg2+ analysis (it could be entitled as ‘’Electrochemical performance of SPE-CSE for Hg2+ analysis’’, with the appropiate subsections) and another one focused on the application of sensor for mercury detection in wastewater (it could be named as ‘’Electrochemical analysis of Hg2+ in wastewater samples’’). In the last case, the table 10 can be moved to this subsection.

-Although the aim of this article does not lie into the development of the electrochemical sensor, I think that the characterization of the device can be included before the electrochemical analysis of Hg2+. For example, the cyclic voltammograms recorded with SPE-PolyL and SPE in the buffer solution without analyte could be included. This could be useful to check the successful electrodeposition of the polymer film on SPE.

-Other relevant point is the preliminary assessment of SPE-PolyL and SPE devices for Hg2+ detection. In this sense, the DPV voltammograms recorded in the buffer solution with the unmodified SPE and SPE modified with the polymer film after their immersion in the mercury complexation solution could be shown before Hg2+ calibration for comparison.

-I would remove the table 1 (line 230), since the relevant information from calibration study (slope, standard deviation values and determination coefficient) could be provided in the body text. On the other hand, Figure 1 could be modified by including a DPV voltammogram recorded with the SPE modified sensor in presence of different Hg2+ concentrations, together with the calibration curve and its corresponding linear regression.

-Consider the decreasing of the number of significant digits of the results exposed in the tables 2, 3, 7, 8, 9 and 10 to three. For example, Xaverage for the added concentration of 20 µg/L exposed in the Table 3 could be shorted to 22.9 µg/L and Sr and r should be maintained as ‘’3.44 µg/L’’ and ‘’9.64 µg/L’’ respectively.

-Consider to include the expressions employed for LOD and LOQ calculations (lines 233-237).

-More information can be added to the recovery section (lines 270-271), such as the definition of recovery and its importance in the analytical field, among others. Furthermore, consider to include some information about the repeatability of the recovery data if some replicates were performed in this study.

-The part related to the statistical analysis of data from robustness part (lines 273-324) is really interesting. I only would like to point out the terms chosen to define the replicates. For me, the term ‘’sample’’ reported in the table 8 is not accurate. Consider to use another term for clarification, such as ‘’replicate’’.

-Some aspects regarding the selectivity of the mercury determination could be included in this part (for example, the addition of other heavy metal complexation solutions together with the mercury one in the stripping step and after then, the recording of a differential pulse voltammogram under the analytical conditions previously shown in the article). This study could be useful to check if their corresponding oxidation potentials are located at different values and hence, mercury could be determined in presence of other metals.

Discussion

-I think that it is not necessary to divide the discussion section (lines 332-382) into different subsections.

-Concerning the robustness method, (lines 353-361), I think that a deep statistical explanation from the data exposed in Tables 8 and 9 could be provided for fully understanding of the results. For instance, the discussion of the results shown in these tables using the relative and absolute effects could be fuitful.

Conclusions

-The conclusions derived from robustness method (lines 388-391) could be rewriten. Here I am including one example: ‘’According to the robustness tests, pH and acetate buffer concentration does not influence the method in the  range studied (±10% variation), while reaction time has a strong effect in the procedure. Hence, the last parameter should be carefully considered to obtain adequate results’’.

-The last sentence (lines 396-398) can be moved to the beginning of the section. On the other hand, no specific values are necessary, since they were specified in the abstract and results sections. Consider to use other expressions, such as ‘’wide linear range’’, low LOD and LOQ values’’, ‘’suitable repeatability and intermediate precission values’’, etc. 

Author Response

Dear anonymous reviewer,

We want to thank you for your time and effort spent to help us to improve our work and the manuscript. Your help was much appreciated. Please find below the answers to your comments and observations:

Introduction

  1. At the beginning, “mercury” word was frequently used. To avoid repetitions, | think that other words, such as " this element", "this metal" or other ones can be employed.

Thank you for the suggestion. We made the proper modifications in the manuscript.

  1. I understand that the use of SPE-modified materials for the detection of mercury were reported in lines 86-89, but the meaning is unclear. | believe that this part can be reorganized as follows: "Several chemically modified electrodes with suitable characteristics, such as low-cost, simplicity of use, high selectivity and sensitivity were developed. Moreover, they can be included in portable devices, allowing in-situ monitoring of analyzed samples”.

Thank you for the suggestion. We made the proper modifications in the manuscript.

  1. Some aspects concerning the use of polymers deposited on SPE for the monitorization of heavy metals could be mentioned after the exposition of screen printed sensors for mercury detection (lines 109-117) and before the purpose of the paper (lines 118-121). Afterwards, the employment of azulene-based sensors could be mentioned as a suitable option to monitorize mercury, based on some reports cited below in the paper. This is an example of what | mean: “Chemically modified electrodes based on polymeric films were also tested for heavy metals monitoring in water samples. Particularly, azulene-based sensors were tested for some heavy metals detection, obtaining adequate results. Therefore, their employment for mercury monitoring may be promoted”. This part of the paragraph can be linked with the one included in lines 118-121. Some research works should be cited in this part. Here | am including some examples after performing a brief literature search:

*A high-performance electrochemical sensor for the determination of based on conductive dopamine polymer doped polypyrrole hydrogel, https://doi.org/10.1016/j.jelechem.2021.115815.

*Novel nano-engineered environmental sensor based on polymelamine/graphitic-carbon nitride nanohybrid material for sensitive and simultaneous monitoring of toxic heavy metals,

https://doi.org/10.1016/j.jhazmat.2021.126267.

*Azulene-ethylenediaminetetraacetic acid: A versatile molecule for colorimetric and electrochemical sensors for metal ions, https://doi.org/10.1016/j.electacta.2018.01.059.

Thank you for the suggestion. We made the proper modifications in the manuscript.

Experimental

  1. From my point of view, the experimental part is not well-organized.| think that it would be better to include all reagents within one subsection entitled “Reagents”, the instrumentation within another one entitled “Instrumentation” and the procedures in different subsections (electrodeposition of poly-L on SPE, electrochemical analysis of Hg2+ in buffer and wastewater samples, sampling of wastewater, among others).

Thank you for the observation. We rearranged the Experimental section as you suggested.

  1. The caption of the figure 1 refers the poly-L structure, but the figure includes the monomer (L). Moreover, | think that this figure can be moved to the end of the introduction section.

You are right. The structure is for the monomer. We made the proper changes and place the figure at the end of the Introduction section.

  1. I understand that DPV was employed to check if the electrochemical pretreatment was done successfully (lines 168-169). If | am right, it would be better to modify this part as follows: "before analysis, a differential pulse voltammogram in free analyte solution was recorded".

We added the suggested modification.

  1. The instrumental cyclic voltammetry conditions exposed in lines 170-172 are not clear. It would be more adequate to rewrite this sentence as follows: "a cyclic voltammogram from -0.5 V to 1.2 V was recorded at 24 mV/s". Hence, the last part of this paragraph (lines 170-172) could be removed.

It was an oversight on our part. The experimental conditions belong to DPV, not CV. To be clear, we replaced CV with DPV.

  1. Consider to modify this expression "using voltammetry in differential pulse" (line 182) for this one: "using differential pulse voltammetry (DPV) under the following conditions". I also recommend to remove the expression "Differential Pulse Voltammetry (DPV) conditions” (lines 182-183).

We appreciate your help. The suggested modification were performed in the manuscript.

Results

  1. I think that the results section is not well-presented. In my opinion, it could be divided into two major subsections: one of them focused on the analytical performance of the electrochemical device for Hg2+ analysis (it could be entitled as “Electrochemical performance of SPE-CSE for Hg2+ analysis”, with the appropriate subsections) and another one focused on the application of sensor for mercury detection in wastewater (it could be named as “Electrochemical analysis of Hg2+ in wastewater samples”). In the last case, the table 10 can be moved to this subsection.

Thank you for this suggestion, we make the changes, the results section is now divided in two parts. We moved table 10 from discussion section to results section.

  1. 10. Although the aim of this article does not lie into the development of the electrochemical sensor, | think that the characterization of the device can be included before the electrochemical analysis of Hg2+. For example, the cyclic voltammograms recorded with SPE-PolyL and SPE in the buffer solution without analyte could be included. This could be useful to check the successful electrodeposition of the polymer film on SPE.

Thank you for the suggestion. We tried to focus on validation of an electrochemical procedure. The required details were already been published on glassy carbon electrodes. We performed these types of measurements in order to be sure that the electrodes were modified. Thus, in order to maintain the main purpose of the paper, we added in the experimental section a paragraph regarding the grown of polyL film on a SPE electrode (Figure S1 from supplementary information) and its transfer into free monomer electrolyte (Figure S2 from supplementary information) which emphasis the electrode modification by maintaining its electroactivity.

  1. Other relevant point is the preliminary assessment of SPE-PolyL and SPE devices for Hg2+ detection. In this sense, the DPV voltammograms recorded in the buffer solution with the unmodified SPE and SPE modified with the polymer film after their immersion in the mercury complexation solution could be shown before Hg2+ calibration for comparison.

From our previous researches, we did not find any answer towards mercury ions of the bare electrode by using open circuit accumulation procedure. However, we added in the supplementary material a comparison between SPE-polyL before and after complexation in which the response toward mercury ions could be seen (Figure S3 from supplementary information).

  1. I would remove the table 1 (line 230), since the relevant information from calibration study (slope, standard deviation values and determination coefficient) could be provided in the body text.

Thank you for the suggestion. We removed table 1 and we provided the comments in the body text.

On the other hand, Figure 1 could be modified by including a DPV voltammogram recorded with the SPE modified sensor in presence of different Hg2+ concentrations, together with the calibration curve and its corresponding linear regression.

The DPV voltamograms recorded at different concentration is make the subject of a Invention patent application OSIM (Romanian State Office for Patents and Brands) no. A00464/29.07.2022. Method for obtaining a screen-printed electrode modified with polymeric films and an electrochemical procedure for field determination of mercury concentration in wastewater.

  1. Consider the decreasing of the number of significant digits of the results exposed in the tables 2, 3, 7, 8, 9 and 10 to three. For example, Xaverage for the added concentration of 20 μg/L exposed in the Table 3 could be shorted to 22.9 μg/L and Sr and r should be maintained as “3.44 μg/L” and “9.64 μg/L” respectively.

Thank you for this suggestion, we make the changes in tables 2, 3, 7, 8, 9 and 10, which becomes 1, 2, 6, 7, 8 and 9 after we deleted table 1.   We change also in table 10 (results on wastewater samples).

  1. Consider to include the expressions employed for LOD and LOQ calculations (lines 233-237).

We add in the text at 3.1.2. Subsection the equations used for LOD and LOQ.

  1. More information can be added to the recovery section (lines 270-271), such as the definition of recovery and its importance in the analytical field, among others. Furthermore, consider to include some information about the repeatability of the recovery data if some replicates were performed in this study.

We add in the text at 3.1.7. Subsection definition of the recovery, equation and how was estimated in the study. Also, we add the range of recovery for each concentration.

  1. The part related to the statistical analysis of data from robustness part (lines 273-324) is really interesting. I only would like to point out the terms chosen to define the replicates. For me, the term “sample” reported in the table 8 is not accurate. Consider to use another term for clarification, such as “replicate”.

Thank you for the suggestion. We make the proper adjustment.

  1. Some aspects regarding the selectivity of the mercury determination could be included in this part (for example, the addition of other heavy metal complexation solutions together with the mercury one in the stripping step and after then, the recording of a differential pulse voltammogram under the analytical conditions previously shown in the article). This study could be useful to check if their corresponding oxidation potentials are located at different values and hence, mercury could be determined in presence of other metals.

You are right. The selectivity of the polyL material was previously checked and discussed in our previous paper [reference 45 from the manuscript]. Also, we could not introduce others metal ions in the stripping step since this process occurs in clean metal free electrolyte. However, as discussed more fully in our previous paper [reference 45 from the manuscript] we also perform the influence of most common polluted metal ions on the mercury ions stripping response using SPE-polyL electrodes. As it can be seen from figure S4 (see supplementary information) the mercury ions stripping peak current  remains at 90% from the initial height even when 5 mass equivalents of interfents ions I (I: (Zn(II), Cd(II), Pb(II), Ni(II), Co(III), and Cu(II)) are added.

We introduce this information to 3.1.6. Subsection of the manuscript.

Discussion

  1. I think that it is not necessary to divide the discussion section (lines 332-382) into different subsections.

We appreciate your help. The suggested modification were performed in the manuscript.

Concerning the robustness method, (lines 353-361), | think that a deep statistical explanation from the data exposed in Tables 8 and 9 could be provided for fully understanding of the results. For instance, the discussion of the results shown in these tables using the relative and absolute effects could be fruitful.

We add supplementary information on both results and discussion sections regarding robustness evaluation.

Conclusions

  1. The conclusions derived from robustness method (lines 388-391) could be rewriten. Here | am including one example: “According to the robustness tests, pH and acetate buffer concentration does not influence the method in the range studied (+10% variation), while reaction time has a strong effect in the procedure. Hence, the last parameter should be carefully considered to obtain adequate results”.

Thank you for this suggestion, we make the changes.

  1. The last sentence (lines 396-398) can be moved to the beginning of the section. On the other hand, no specific values are necessary, since they were specified in the abstract and results sections. Consider to use other expressions, such as “wide linear range”, low LOD and LOQ values”, “suitable repeatability and intermediate precission values”, etc.

Thank you for this suggestion, we make the changes.

Reviewer 2 Report

In this study, the validation of an electrochemical procedure for using of a modified carbon screen-printed electrode (SPE) with a complexing polymeric film (polyL) for Hg2+ ions determination in wastewater samples. The ECS-polyL electrode presents a linearity in the range of 20 μg/L to 150 μg/L, with LOD = 6 μg/L, LOQ = 20 μg/L and an average measurement uncertainty of 26% of mercury ions. The manuscript presents interesting results, which are relatively well organized and systematized, but the novelty and economic impact of this study should be highlighted more. In my opinion, this manuscript should be published after minor revision.

Here is a list of my general comments:

·         The novelty, economic impact and practical applicability of this study should be highlighted more.

·         Define abbreviations at first mention. Abbreviations should be defined at first mention and then through the text use only the abbreviations not the full name and use the same abbreviation through the manuscript.

·         Newer references should be included in the introduction and the discussion part if possible. From 49 references, only 23 are from last five years. It is better that there be at least 50% references from last five years.

·         Specific comments:

o    Line 19: Define abbreviation ECS in the Abstract.

o    Lines 56-58: Give the full name of the term and then give the abbreviation in the brackets.

o    Line 119: The abbreviation for screen-printed electrode is already introduced (line 75), so use it in further text.

o    Line 125: The abbreviation for 2,2′-(ethane-1,2-diylbis((2-(azulen-2-ylamino)-2-oxoethylazanediyl)diacetic acid is already introduced (line 120), so use it in further text.

o    Line 126: The abbreviation for screen-printed electrode is already introduced (line 75), so use it in further text.

o    Line 129: The abbreviation for 2,2′-(ethane-1,2-diylbis((2-(azulen-2-ylamino)-2-oxoethylazanediyl)diacetic acid is already introduced (line 120), so use it in further text.

o    Line 131: The abbreviation for screen-printed electrode is already introduced (line 75), so use it in further text.

o    Line 157: What is S in the ECS abbreviation of electrochemical cell?

o    Line 166: Define abbreviation for cyclic voltammetry (CV) here, not in line 170.

o    Line 191: The abbreviation for LOD is already introduced (line 114).

o    Line 233: The abbreviations for LOD and LOQ are already introduced (line 114).

o    Line 363: The abbreviation for screen-printed electrode is already introduced (line 75), so use it in further text (line 75).

o    Line 384: The abbreviation for screen-printed electrode is already introduced (line 75), so use it in further text.

o    Line 386: The abbreviations for detection limit and quantification limit are already introduced (line 114). Use only abbreviation.

o    Line 395: The abbreviation for screen-printed electrode is already introduced (line 75), so use it in further text (line 75).

Author Response

Dear anonymous reviewer,

We want to thank you for your time and effort spent to help us to improve our work and the manuscript. Your help was much appreciated. Please find below the answers to your comments and observations:

  1. The novelty, economic impact and practical applicability of this study should be highlighted more.

Thank you for the suggestion. We introduced a paragraph in Introduction section to emphasis your suggestion as follows: “In addition, the present work intends to make a step further into the development of an electrochemical portable, with low running costs and easy to operate system based on modified SPE for mercury detection. “

  1. Define abbreviations at first mention. Abbreviations should be defined at first mention and then through the text use only the abbreviations not the full name and use the same abbreviation through the manuscript.

You are right. We check the manuscript and perform the proper changes.

  1. Newer references should be included in the introduction and the discussion part if possible. From 49 references, only 23 are from last five years. It is better that there be at least 50% references from last five years.

Thank you for your comment. We replace some references and add supplementary ones. There are now 28 references in the range 2018-2022 from 52 references represented 53.8% from the last five years. Some of the references are related with the validation procedure, represented standards, books or articles older than 5 years.

Specific comments:

  1. Line 19: Define abbreviation ECS in the Abstract.

Thank you for the observation. It was an oversight on our part. We replace ECS with SPE

  1. Lines 56-58: Give the full name of the term and then give the abbreviation in the brackets.

You are right. For the techniques that are not used further in the manuscript, we let the full name.

Line 119: The abbreviation for screen-printed electrode is already introduced (line 75), so use it in further text.

Thank you for your recommendation. We use the abbreviation for screen-printed electrode further in the manuscript.

Line 125: The abbreviation for 2,2'-(ethane-1,2-diylbis((2-(azulen-2-ylamino)-2-oxoethylazanediyl)diacetic acid is already introduced (line 120), so use it in further text.

We use the abbreviation as you recommended. 

Line 126: The abbreviation for screen-printed electrode is already introduced (line 75), so use it in further text.

We make the proper adjustment.

Line 129: The abbreviation for 2,2'-(ethane-1,2-diylbis((2-(azulen-2-ylamino)-2-

oxoethylazanediyl)diacetic acid is already introduced (line 120), so use it in further text.

We make the proper adjustment

Line 131: The abbreviation for screen-printed electrode is already introduced (line 75), so use it in further text.

We make the proper adjustment.

Line 157: What is S in the ECS abbreviation of electrochemical cell?

Thank you for the observation. It was a misleading. It was intended to be EC.

Line 166: Define abbreviation for cyclic voltammetry (CV) here, not in line 170.

We make the proper adjustment.

Line 191: The abbreviation for LOD is already introduced (line 114).

We make the proper adjustment.

Line 233: The abbreviations for LOD and LOQ are already introduced (line 114).

We make the proper adjustment.

Line 363: The abbreviation for screen-printed electrode is already introduced (line 75), so use it in further text (line 75).

We make the proper adjustment.

Line 384: The abbreviation for screen-printed electrode is already introduced (line 75), so use it in further text.

We make the proper adjustment.

Line 386: The abbreviations for detection limit and quantification limit are already introduced (line 114). Use only abbreviation.

We make the proper adjustment.

Line 395: The abbreviation for screen-printed electrode is already introduced (line 75), so use it in further text (line 75).

We make the proper adjustment.